# Surgical Site-Released Tissue Is Potent to Generate Bone onto TCP and PCL-TCP Scaffolds In Vitro

**DOI:** 10.3390/ijms242115877

**Published:** 2023-11-01

**Authors:** Emely Rehage, Andrea Sowislok, André Busch, Eleftherios Papaeleftheriou, Melissa Jansen, Marcus Jäger

**Affiliations:** 1Chair of Orthopaedics and Trauma Surgery, University of Duisburg-Essen, 45147 Essen, Germany; emelyre0906@gmail.com (E.R.); andrea.sowislok@uni-due.de (A.S.); 2Department of Orthopaedics, Trauma and Reconstructive Surgery, Katholisches Klinikum Essen Philippus, 45355 Essen, Germany; 3Department of Orthopaedics, Trauma and Reconstructive Surgery, St. Marien-Hospital Mülheim an der Ruhr, 45468 Mülheim an der Ruhr, Germany; e.papaeleftheriou@contilia.de; 4Institute of Cognitive Science, University of Osnabrück, 49090 Osnabrück, Germany; mejansen@uni-osnabrueck.de

**Keywords:** surgical site-released tissue, bone healing, tissue collector, scaffold, tricalciumphosphate

## Abstract

There is evidence that surgical site tissue (SSRT) released during orthopedic surgery has a strong mesenchymal regenerative potential. Some data also suggest that this tissue may activate synthetic or natural bone substitute materials and can thus upgrade its osteopromoting properties. In this comparative in vitro study, we investigate the composition of SSRT during total hip replacement (*n* = 20) harvested using a surgical suction handle. In addition, the osteopromoting effect of the cells isolated from SSRT is elucidated when incubated with porous beta-tricalcium phosphate (β-TCP) or 80% medical-grade poly-ε-caprolactone (PCL)/20% TCP composite material. We identified multiple growth factors and cytokines with significantly higher levels of PDGF and VEGF in SSRT compared to peripheral blood. The overall number of MSC was 0.09 ± 0.12‰ per gram of SSRT. A three-lineage specific differentiation was possible in all cases. PCL-TCP cultures showed a higher cell density and cell viability compared to TCP after 6 weeks in vitro. Moreover, PCL-TCP cultures showed a higher osteocalcin expression but no significant differences in osteopontin and collagen I synthesis. We could demonstrate the high regenerative potential from SSRT harvested under vacuum in a PMMA filter device. The in vitro data suggest advantages in cytocompatibility for the PCL-TCP composite compared to TCP alone.

## 1. Introduction

Delayed or failed bone healing is a common complication in orthopedic and trauma surgery. Due to the different causes (trauma, tumor, infection, and osteonecrosis), the treatment is challenging and includes both biological and biomechanical aspects. Since bone healing is disturbed by macro- and micromovements of the surrounding tissue, the affected area requires external or internal stabilization. Moreover, sufficient blood supply of the affected area is a prerequisite for local nutrition, allowing for the migration of different cell types, and thus, bone regeneration. Systemic metabolic disorders such as diabetes mellitus, occlusive arterial disease, microangiopathy, storage disorders, and disruption of the immune system (neoplasm, chemotherapy, AIDS, etc.) can also affect bone healing.

Under physiological conditions, bone healing goes through distinct characteristic phases (Figure 1a). Initially, tissue damage (trauma and surgery) causes bleeding, followed by the activation of coagulation and the complement cascade [1]. Moreover, thrombocytes and other cells release their cytokines as well as inflammatory factors (prostaglandins and leukotrienes). Attracted by chemotaxis, macrophages and other immunocompetent cells are predominantly present in the first stages of bone healing, removing necrotic tissue and releasing cytokines and growth factors. The corresponding clinical feature is an acute, aseptic inflammation (pain, swelling, and hematoma). Later on, attracted by cytokines, local growth factors and other molecular mediators, mesenchymal stromal cells, migrate into the osseous defect area. The chemo-taxis induced accumulation of these MSCs and other cells (lymphocytes, granulocytes, and hematopoietic cells) are a prerequisite for osteoblast differentiation and the formation of new bone (callus) [2]. At least, bone healing is orchestrated by an appropriate microenvironment including a mixture of growth factors and hormones such as bone morphogenetic proteins (BMPs), vascular endothelial-derived growth factor (VEGF), platelet-derived growth factor (PDGF), transforming-growth factor β1 (TGF β1), osteogenic growth peptide (OGP), and fibroblastic growth factor (FGF) (Table 1) [3,4].

Knowing the principal mechanisms and key factors of in vivo bone healing, we assume that tissue, which is damaged and released during orthopedic surgery, also includes essential stimuli to induce local bone regeneration. Thus, some data from the literature suggest collecting and applying this tissue to promote bone healing with and without bone substitute materials, such as TCP or allograft material [5,6,7,8].
Figure 1(**a**,**b**) The flow chart shows the different stages of bone healing according to [4,9,10] (**a**). At the very beginning of this cascade, surgical site-released tissue is collected with a vacuum using a PMMA-based suction handle. The system includes an internal filter retaining small tissue fragments as well as clotted blood and thus the biological information to initiated bone healing. Here, bone substitute materials are coated and activated via these autologous tissue components (**b**).
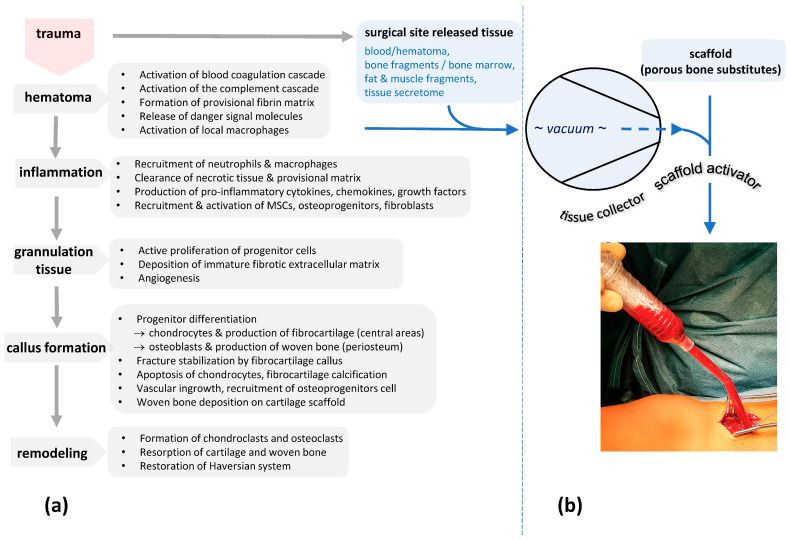


In this study, we investigated if the surgical site-released tissue components are competent to initiate bone regeneration in vitro when incubated onto two different scaffolds qualified as bone void fillers.

## 2. Results

### 2.1. Yield of Mononuclear Cells (MNCs), Proliferation Potential, and Stemness Character

The average weight of the collected surgical site-released tissue from all 20 samples was 17.7 g ± 6.9 g. Mononuclear cells (MNCs) could be isolated and cultured in all cases, yielding a mean baseline number of 0.87 ± 0.79 × 10^10^ cells per sample or 0.054 ± 0.026 × 10^10^/g per gram of collected tissue. After gradient centrifugation, the number of MNCs decreased by a factor of 10^2^ to 19.8 ± 13.8 × 10^7^ or 1.09 ± 0.42 × 10^7^/g. The proliferation potential determined as generation time was comparable in P1 and P2, with 8.93 ± 4.1 days in P1 and 11.6 ± 5.02 days in P2. A colony-forming unit (CFU) assay with MNCs of all samples revealed an average value of 9.42 ± 10.18 ×10^6^ MNCs. From these results, the theoretical number of potential MSCs in the samples was calculated, resulting in an average number of 1.36 ± 1.96‰ MSCs per sample corresponding to 0.09 ± 0.12‰/g per gram of tissue.

### 2.2. Differentiation Potential

The presence of MSCs was controlled based on the expression of the typical markers via flow cytometry. Here, the cells of all samples showed a significant expression of mesenchymal stromal cell markers (CD73+, CD 90+, and CD105+) and no hematopoietic cell markers (CD31−, CD34−, and CD45−).

The three-lineage specific differentiation was possible in all cases, as shown by the cytochemical staining of calcium (alizarin red; osteoblasts), glycosaminoglycans (alcian blue; chondroblasts), and triglycerides (oil red; adipoblasts).

### 2.3. Analysis of the Growth Behavior of MNCs on Bone Substitute Materials

#### 2.3.1. Light Microscopic Evaluation during the Culture Period

The isolated MNCs were seeded in the center of the biomaterial ring prior to gradient centrifugation and examined weekly via light microscopy during the culture period. In general, a tendency for higher and faster cell growth of MNCs was observed in the PCL-TCP well compared with the β-TCP well (Figure 2). The MNCs grew earlier, after about 2 weeks in the PCL-TCP well, whereas the MNCs in the β-TCP well did not increase significantly in cell number until 1–2 weeks later. The optical scoring system showed that the β-TCP group exhibited cell growth in 52% of all samples, meaning that in 14 out of 27 samples, one of the categories +/++/+++/++++ was fulfilled and that cell growth was generally occurring in the well. Particularly strong cell growth was observed in the β-TCP wells in 26% of cases, meaning that in 7 out of 27 cases, the cell growth could be classified as +++ or ++++. In comparison, the PCL-TCP group showed cell growth in 70% of all samples, so that it could be evaluated that general cell growth, according to category +/++/+++ or ++++, was present in 19 out of 27 samples. Strong cell growth, corresponding to the categories +++ or ++++, was seen in 59% of cases (in 16 out of 27 samples) for the PCL-TCP group.

#### 2.3.2. Quantification of Cell Proliferation via ATP Assay

Cell proliferation on both biomaterials was assessed using an adenosine triphosphate (ATP) assay after 2, 4, and 6 weeks of culture. ATP concentrations correlated linearly with cell numbers. At two weeks, the samples with and without RBC lysis differed significantly with the samples undergoing RBC lysis, exerting lower concentration levels of ATP (t (20.543) = 3.3989, CI = [1.01, 4.22], and *p* = 0.0028). After 4 weeks, however, no noticeable differences could be detected between the samples with and without RBC lysis (t (7.9432) = 0.68142, CI = [−1.92, 3.54], and *p* = 0.515). Therefore, both conditions were included in the statistics from a time point of 4 weeks. Visualization of the obtained ATP values showed a general increase in ATP concentration during the six-week culture period in all groups. When comparing the two bone graft substitutes, a higher cell density was observed in the samples cultured with PCL-TCP, as shown by higher ATP levels of the isolated ring and the surrounding area (Figure 3).

Using a linear model that yielded an estimate of 0.5930 (*p* = 0.66440), no main effect was found for the surface variable, so the well and ring materials can be considered equally good and comparable for cell adhesion. No effects of interaction were found. A Student’s *t*-test showed a strongly significant effect for the PCL-TCP well compared to the β-TCP well alone (t (4.4339) = 4.1095 CI = [2.590633], and *p* = 0.00596). A linear model fitted to the data revealed a significant main effect of the substrate in favor of PCL-TCP (t (12) = 5.407, estimate = 5.2285, and *p* < 0.001) when a combination of the well and substrate was considered. This is also underlined by a Student’s *t*-test showing better results for the PCL-TCP substrate compared to the β-TCP substrate considering the respective wells (t (17.912) = 3.7831 CI = [2.018635], and *p* = 0.0006858). When the effect size of the PCL-TCP ring was considered separately with a linear fixed model, a relevant positive trend in favor of the PCL material was found (*p* = 0.048591). The mean values in the *t*-test for the PCL-TCP ring and the β-TCP ring showed a slightly higher mean of 9.833961 for PCL-TCP than for β-TCP with a mean of 7.606905.

#### 2.3.3. Localization and Distribution of Viable Cells on Biomaterials Using MTT Staining

After six weeks in culture, vital, cell-occupied areas on the bone substitutes were stained purple with 3-(4,5-Dimethylthiazol-2-yl)-2,5-diphenyltetrazoliumbromid (MTT). For β-TCP rings, only 32% of all rings (13 of 41) showed positive staining and had single, circular cell colonies with a few confluent sites. In comparison, 56% of all PCL-TCP rings (23 of 41) stained positive, with the cells growing more widely distributed rather than forming single colonies. Cell borders were less visible than in the β-TCP material and staining was more widely distributed (Figure 4).

#### 2.3.4. Visualization of Cells on Both Biomaterials Using Fluorescence Microscopy and SEM

After six weeks in culture, biomaterials stained positive with MTT were examined via fluorescence- and scanning electron microscopy (SEM), which confirmed cell growth on both biomaterials. Visualization of the cytoskeleton via actin staining revealed biomaterial specific differences in cell morphology in fluorescence micrographs. While the MSCs in the β-TCP ring showed an elongated, spindle-shaped morphology with many thin cytoplasmic extensions (Figure 4a), the morphology of the MSCs in the PCL-TCP ring resembled that of the MSCs in the cell culture flasks. The cells in the PCL-TCP bone substitute had a more planar, broader morphology and grew more homogeneously distributed over the material (Figure 4b). SEM micrographs were consistent with these results. Here, differences in the surface structure of both biomaterials became obvious. While β-TCP rings had an uneven, porous structure, the structure of PCL-TCP appeared smooth (Figure 4e,f).

#### 2.3.5. Secretion of Osteogenic Marker Proteins

Secretion of the osteogenic marker proteins osteocalcin (OCN), osteopontin (OPN), and collagen I (COL I) into the cell culture medium was determined via enzyme-linked immunosorbent assays (ELISAs) during cell cultivation. In general, an increase in the concentration of the measured proteins (OCN, COL I, and OPN) was observed during the cultivation period (Figure 5a–c). Statistical comparison of the two adhesives using Student’s *t*-test showed no significant differences in favor of any adhesive, so that glue and agarose performed equally well in all tests (Table 2). For further statistical analysis, both adhesive conditions were combined, and a time point of five weeks, was chosen for assessments. A one-sided Student’s *t*-test showed significant effects for the higher secretion levels of osteocalcin in the PCL-TCP group compared to the β-TCP group (t (10.208) = 2.9823 *p* = 0.006731). This significant main effect was confirmed via a linear model (t (10.59) = 3.68, estimate = 0.36273, and *p* = 0.00895) (Figure 6a). However, for osteopontin, the linear model revealed no significant differences, meaning that β-TCP and PCL-TCP performed equally well in terms of the amount of osteopontin secretion (Figure 6b). The same applies to the analysis of collagen I. A positive trend in favor of PCL-TCP can be seen in the data visualization (Figure 7c). In addition, Figure 6c shows a positive trend towards higher collagen I values when glue is used as an adhesive instead of agarose.

### 2.4. Biochemical Analysis of the Aspirated Surgical Site-Released Tissue

The harvested SSRT was further analyzed and compared with the patient’s blood in terms of cellular composition, osmolarity, water, growth factors, and fat content. Tissue electrolytes and the total protein content were largely comparable to those of the patient’s blood, except for the potassium, phosphate, and triglyceride levels, which were elevated (Table 3). Potassium and phosphate levels were even 1.7 to 2 times higher than the reference values. Hemoglobin levels in tissue samples were lower than those in the blood and reference values. The water content of the surgical tissue was 49.3% ± 7.2%.

Cytometry flow analysis of the cells isolated from SSRT after Ficoll gradient centrifugation and RBC lysis revealed positive expressions of CD34 (98.21% ± 0.019), CD31 (97.50% ± 0.029), CD44 (83.61% ± 0.143), and HLA-DR (92.14% ± 0.1). Mesenchymal stromal cell markers (CD73+, CD 90+, and CD105+) were positive in only less than 5% of all cells (Figure 8).

A general analysis of growth factor distribution in tissue and plasma samples from one patient via microarray revealed the presence of 39 of the 41 analyzed human growth factors. Furthermore, no significant differences were detectable between the two samples. In both samples, the following growth factor families could be detected (see Figure 9a,b for more detail): Colony-stimulating factors (#2A,3B,3C), stem cell factors (#3L,4A), transforming growth factors (#4B,4C,4D,4E), fibroblast growth factors (#1F,1J,1K,1L), epidermal growth factors (#2D,1H,1I), platelet-derived growth factors (#3H,3I,3J,3F,3G), vascular endothelial growth factors (#4F,4G,4H), insulin-like growth factors (#2K,2L,3A), insulin-like growth factor binding proteins (#2F,2G,2H,2I,2J), and others (#3D,1E,2E,3E,3K,2B,1G).

Within these cytokines we quantified the concentration of VEGF and PDGF for SSRT and peripheral blood. Both factors were significantly increased in SSRT (Figure 9c).

## 3. Discussion

The idea to use surgical site-released tissue to stimulate bone healing is not new [11,12,13,14]. Especially during manipulation on the cancellous bone (e.g., osteotomy, total joint arthroplasty, and osteosynthesis), liberated bone pieces, and its marrow as well as the fat, present an excellent source for bone regeneration. Here, some tissue collectors such as the RIA procedure or the application of BMAC have contributed to reducing autologous bone grafting. However, most systems require an independent surgical approach with its co-morbidities to the patient, which may prolong surgery or incur additional costs [15].

Our data describe the composition of surgical site-released tissue during total hip replacement in patients with advanced osteoarthritis. Here, we found several cytokines within the soluble phase which occur also in peripheral blood, including high amounts of CD34 cells. Therefore, it is not surprising that the soluble phase of SSRT and peripheral blood does not differ significantly in many parameters (Table 3). The increased levels of triglycerides may be a result of collected fat and the increased levels of potassium in SSRT can be explained via a concentration of adhesive platelets and cellular lysis.

As a major result, our data support the hypothesis that the initial hematoma following fracture or during surgery is crucial for the initiation of bone healing [16]. Moreover, this in vitro study as well as previously published data confirms the high regenerative potential of this tissue composite [5]. We found significantly higher PDGF and VEGF concentrations in the SSRT compared to peripheral venous blood (Figure 9). This might also explain the increased bone healing of PCL-TCP/SSRT in a segmental tibial defect in sheep (unpublished data) compared to the controls. However, the first clinical data also demonstrated excellent bone regeneration in TCP/SSRT composites [17].

The interconnections between scaffolds and growth factors in bone regeneration are sophisticated and are the subject of current research and scientific discussion. Especially polymer polycaprolactone (PCL) is a promising candidate to bind growth factors as soluble proteins. This includes not only the ability to use growth factor-loaded PCL scaffolds for a controlled release of proteins [18], e.g., by electrospinning [19], but also the potency of unloaded PCL to bind local growth factors from the adjacent microenvironment such as in cell cultures or body fluids. One example for this phenomenon is VEGF, a factor which is crucial for bone regeneration and neovascularization and showed high levels in our study. Suárez-González et al. incubated PCL scaffolds in modified simulated body fluids and found out that peptide versions of vascular endothelial growth factor (VEGF) and bone morphogenetic protein 2 (BMP2) were bound with efficiencies up to 90% to mineral-coated PCL scaffolds [20].

Moreover, some data suggest that the release of growth factors such as VEGF or PDGF adhered onto PCL is strongly dependent from the solubility and type of the mineral coating [21,22,23,24,25]. In addition, there are several techniques to functionalize PCL-scaffolds for protein binding such as plasma hydrolysis, alkali hydrolysis, or aminolysis. [26,27]. We did not further investigate these effects in our study but are aware that further investigations are required to elucidate the interconnections between TCP or PCL-TCP and its defined proteins (e.g., VEGF, BMPs, or PDGF).

Furthermore, the in vitro data suggest that poly(3-caprolactone) (PCL)/β-tricalcium phosphate (β-TCP) is superior to highly porous β-TCP regarding cellular adherence, growth, and differentiation. These data correspond to Tabatabaei et al. [28], who coated this material with collagen and also confirmed the preclinical data on bone regeneration in a sheep model [29].

The biodegradable polyester poly-ε-caprolactone is an aliphatic semi-crystalline polymer with a relative degradation time of 2–3 years in vivo via hydrolysis of its aliphatic ester linkage [30,31]. In its untreated state, it is relatively hydrophobic compared to TCP with its ion release and shows a rubbery state under physiological temperature [32]. However, surface pretreatment (e.g., via NaOH) and 3D-printing can change the surface behavior, thus allowing cell adhesion (optimal water contact angle for cell adhesion is between the range of 30–70°) [33]. But also, the composite with β-TCP slightly increases the hydrophilic properties of the mixture. This could explain the high adherence and superior growth of MNC onto PCL-TCP in our in vitro study. In contrast, the large surface of highly porous β-TCP promotes ion release and leads to significant changes in the pH value in vitro (in DMEM up to 8.08 after 1440 min), which results in cellular death [34]. At this point, it is hard to transfer the in vitro data of our study to the clinical situation. For the latter, diluting and buffering effects as well as protein coating occur in situ as soon the biomaterial meets body fluid. Here, the tissue collector used in this study (surgical suction handle) might have the advantages anticipating these effects outside the tissue.

The superior properties of PCL scaffolds were also documented by Faia-Torres et al. who found that a surface roughness of Ra~0.9–2.1 μm showed excellent osteogenic differentiation in hMSCs seeded on PCL scaffolds and cultured in a dexamethasone-deprived osteogenic induction medium [35]. We did not quantify the surface parameters (Ra, Rz, Rt, and CA) but the SEM analysis showed a relatively low porosity and a smooth surface of PCL-TCP compared to the highly porous calcium phosphate crystals of TCP.

Also, the increased osteocalcin expression in the PCL-TCP as a typical protein characterizing the differentiated osteoblast corresponds to other investigators. [36,37] and may have a positive effect to bone formation. In addition to other functions, osteocalcin has been found to regulate hydroxyapatite (HA) formation via the acceleration of crystal nucleation and the inhibition of the apatite (0001) plane, suppressing crystal growth perpendicular to this plane [38]. In contrast, we found no significant differences in the collagen I and osteopontin expression between both biomaterials (TCP vs. PCL-TCP).

Furthermore, our experiments showed an inhibitory effect of agarose in the osteopontin and collagen expression in both TCP and PCL-TCP specimens. These findings correspond to other authors who found that the addition of agarose to cell cultures progressively restricts cell spreading, reduces the stress fiber and focal adhesion assembly, but especially impairs the cell-directed assembly of large collagen bundles and remodeling [39,40]. Therefore, it is not surprising that in contrast to cartilage regeneration, agarose-based biomaterials have not been able to establish themselves as bone-guided scaffolds. However, to our knowledge, the inhibitory effects of the agarose to osteopontin expression have not been described so far. The cytocompatibility of the epoxy resin we used for the fixation of the ring was excellent and did not affect cellular growth or the expression of osteogenic markers significantly. In accordance, epoxy resin-based root canal sealers are in clinical application in dentistry for years [41]. Although some data document dose-dependent cytotoxicity and genotoxicity for epoxy resins in vitro [42], other results confirm a relative good cytocompatibility for the most products in its polymerized stage [43]. In our study, we showed that the cultivated cells from SSRT migrated into the 3D micropores of both scaffolds and found qualitative differences in SEM. Although we did not quantify the number of the cells adhered to the scaffold directly, we measured the released ATP concentration as indirect evidence for cell migration.

Moreover, long-term cultures may be problematic for nutrients to supply the cells through the 3D network of the scaffolds via diffusion. Here, also the progressive adhesion of protein layers onto the surface may clog and block small pores, and thus, inhibit the diffusion of nutrition. For further studies, these disadvantages may be solved using bioreactors.

But, how can components of the surgical site-released tissue initiate bone formation? The answer to this question remains hypothetical. SSRT concentrates the soluble and solid components of mechanical-traumatized tissue. It consists of a highly activated biological composite, requesting signal-mediated assistance from different systems of the body (immune system, regeneration system, and senso-motoric system) to prevent further damage and loss of tissue, and thus, loss of function [44]. These signals derived from a mixture of burst platelets, which release its cytokines and growth factors (a) from factors of the activated coagulation system (b) as well as from the activated complement (c) and immune system (d) [45]. Some signals may also be released by the ripped tissue fragments (e) such as soft tissue (fat, muscle, fascia, and periosteum), as well as bone and its marrow [46,47,48,49,50,51,52]. These effects are accompanied by local electrolyte displacement such as increased extracellular K^+^ due to cellular death and result in at least an aseptic inflammation as shown in Figure 1. Since the ability to adhere onto plastic surfaces is a criterion for mesenchymal stroma cells—the progenitors of osteoblasts—the PMMA filter of the harvesting device may also have a potential to increase the number of MSCs, and thus, promote osteoblast differentiation. As published previously, the cells isolated from SSRT have shown great osteopromotion in vitro [5,53]. This hypothesis is also supported by Groven et al. who used the same tissue collector device (BoneFlo^®^) and showed exclusively expressed osteogenic miRNAs as well as overall enhanced osteogenic marker gene expression in SSRT [54].

Therefore, we believe that the application of SSRT into a local bone defect with all its inflammatory and regenerative stimuli is a promising candidate to initiate bone formation, even in non-unions or critical-size bone defects. However, we are aware that the transformation of in vitro data to the in vivo situation is limited. Especially, further studies are required to obtain a more comprehensive quantitative biochemical and kinetic profile of those factors and cytokines which are predominantly involved in bone formation.

By and large, our in vitro data showed a superiority of PCL-TCP scaffolds compared to TCP regarding bone regeneration. In the future, it might be useful to compare also pure PCL as a control, in order to document the PCL-related in vitro effects under the scenario described in this study. However, it is not clear if these results can be transferred to the in vivo situation. Here, further investigations are required to elucidate the clinical impact of these effects.

## 4. Material and Methods

### 4.1. Patients

Following a prospective design, the patient cohort consisted of 20 patients (15 females, five males, mean age of 68.4 ± 11.6 years) who were scheduled for elective total hip replacement due to advanced osteoarthrosis. The study protocol was approved by the local ethical committee (No 22-10899-BO, ethical board of the medical faculty of the University Duisburg-Essen, Germany). All individuals had given their written informed consent prior to surgery. There were no specific exclusion criteria. A total of five patients were recruited for biochemical analysis of the surgical site-released tissue (four females, one male, mean age 71.2 ± 6.1 years). Inclusion criteria and prerequisites were the same as mentioned above.

### 4.2. Sample Collection Using a Surgical Vacuum Suction Device

An antero-lateral approach to the hip joint was performed according to Harding–Bauer [55,56]. During surgery, the internal filter of a conventional surgical vacuum suction handle (BoneFlo^®^, TissueFlow, Essen, Germany) was used for 20 min to aspirate and collect the patient’s surgical site-released tissue (blood, fragments of bone/bone marrow, muscle, and fat). The complete suction handle was then transported to the laboratory under sterile, cool conditions at 4 °C for further processing. In addition, 7.5 mL of intravenous blood was drawn during the procedure to compare the surgical site tissue with the patient’s peripheral blood.

### 4.3. Isolation and Cultivation of Cells

Isolation and cultivation of mononuclear cells (MNCs) from autologous tissue harvested in the BoneFlo^®^ system was performed as previously described [53]. Briefly, the tissue was incubated with 300 units of streptokinase (Sigma-Aldrich, Deisenhofen, Germany) in Dulbecco’s phosphate-buffered saline (DPBS) (gibco, ThermoFisher Scientific, Waltham, MA, USA) for 15 min at room temperature (RT). The sample was then washed with DPBS containing 2% fetal calf serum (FCS) (Biochrom AG, Berlin, Germany), then filtered and centrifuged at 300× *g* at RT for 10 min. The cell number of the resulting MNC pellet was determined and the cells were seeded for a colony-forming unit (CFU) assay and for biomaterial testing (see Section 4.4 and Section 4.5). After Ficoll density gradient centrifugation (Ficoll Paque Plus, Cytiva, Uppsala, Sweden), the remaining cells were cultured in T25 tissue flasks at 37 °C and 5% (*v*/*v*) CO_2_ in low-glucose DMEM (gibco, ThermoFisher) supplemented with 10% (*v*/*v*) FCS and penicillin–streptomycin 5000 U/mL (gibco, ThermoFisher). The medium was changed twice a week every third day. At 80% confluence, the adherent cells were detached with accutase (StemPro, gibco, Waltham, MA, USA), then counted and seeded at a density of 3.5 × 10^3^ cells per cm^2^ in T75 tissue flasks. After the third passage, the cells underwent flow cytometric analysis (Section 4.9) and were seeded for trilineage differentiation (adipogenic, chondrogenic, and osteogenic) (Section 4.10).

### 4.4. Colony-Forming Unit (CFU) Assay

The colony-forming unit assay was performed as previously described using crystal violet (SERVA Electrophoresis, Heidelberg, Germany) in 20% (*v*/*v*) methanol (J.T. Baker, Gliwice, Poland) [53]. MNCs (see Section 4.3) were cultured in duplicates in 6-well plates in supplemented DMEM at the following cell densities: 5 × 10^6^, 10 × 10^6^, and 20 × 10^6^.

### 4.5. Biomaterials and Cell Culture

Cylindrical synthetic bone grafts made of ß-tricalcium phosphate (ß-TCP) (Cerasorb M^®^, Curasan AG, Kleinostheim, Germany) and tubular composite scaffolds made of 80% medical-grade poly-ε-caprolactone (PCL) and 20% ß-TCP (mPCL-TCP) (Osteopore Pty Ltd., Singapore) were used in this study (Figure 10). Further characteristics of the composite material can be found in the literature [57,58,59,60].

The specimen had an outer diameter of 21 mm, a wall thickness of 3 mm, and a height of 50 mm (ß-TCP) and 100 mm (mPCL-TCP). The mPCL-TCP scaffolds were fabricated using the Fused Deposition Molding technique. Rings with a height of 5 mm were cut from these cylinders. β-TCP rings were sterilized by autoclaving and PCL-TCP rings were γ-sterilized (25 kGy ± 10%). Under a laminar flow bench, the rings were mounted into a TC-treated 6-well plate (Falcon, Taufkirchen, Germany) and fixed using either 2% agarose in DMEM or a 2-component epoxy resin adhesive (UHU, Bühl, Germany). The cells obtained in Section 4.3 were seeded in a density of 40 × 10^6^ into the center of the ring. The following conditions were used in duplicate per patient to investigate the ingrowth behavior of MNCs into both biomaterials: (i) cells only; (ii) β-TCP; (iii) PCL-TCP; (iv) glue (dual-component epoxy adhesive, Fa. Uhu GmbH & Co.KG, Bühl, Germany)/agarose only. The well plates were cultured at 37 °C and 5% (*v*/*v*) CO_2_ in supplemented DMEM for 6 weeks. The medium was changed twice a week every third day. During cultivation, the cells were microscoped once a week to evaluate the growth behavior. The following categories were used for the light microscopic evaluation: (−), no cell growth; (+), single cells or a small colony; (++), several smaller colonies or very few larger colonies; (+++), many large colonies; and (++++), dense cell lawn. The data were calculated from the microscopic results, with +/++ summarized as low cell growth and +++/++++ as high cell growth.

### 4.6. Cell Viability—ATP Assay and MTT Staining

Cell viability was quantified via an adenosine triphosphate (ATP) assay (CellTiter-Glo^®^, Promega, Madison, WI, USA). The cells seeded on rings mounted on a 12-well plate with 2% agarose in DMEM (conditions i–iii, Section 4.5, one per time point) were treated with a red blood cell lysing buffer (Hybrid-Max, Sigma, St. Louis, MI, USA) for 1 min at RT to lyse the erythrocytes before the assay. After washing with DMEM, the assay was performed according to the manufacturer’s instructions after 2, 4, and 6 weeks in culture. The luminescence was measured using a Tecan Reader infinite 200 pro (Tecan Group AG, Männedorf, Switzerland) (integration time 2000 ms).

The staining of viable cells for microscopic visualization was performed via 3-(4,5-Dimethylthiazol-2-yl)-2,5-diphenyltetrazoliumbromid (MTT) (Sigma Aldrich, St. Louis, MI, USA) staining using rings from Section 4.5 after 6 weeks in culture. The medium was aspirated and the cells were washed once with DPBS and incubated with 3 mL of 5 mg/mL MTT in DPBS for 4 h at 37 °C and 5% CO_2_. The cells were then washed with DPBS and microscoped using a VHX 7000 digital microscope (Keyence, Neu-Isenburg, Germany).

### 4.7. Secretion of Osteogenic Marker Proteins

To determine the osteogenic markers, osteopontin, osteocalcin, and pro-collagen Iα, the cell culture supernatant was collected once a week during the 6-week culture period and stored at −80 °C until use. Enzyme-linked immunosorbent assays (ELISAs) (human Duoset ELISA kit, R&D Systems, Bio Techne, Minneapolis, MN, USA) were then performed using the supernatants at a dilution of 1:10 for osteocalcin (1:50; pure for collagen and pure for osteopontin) according to the manufacturer’s protocol and measured at 450 nm using a Multiscan Ascent reader (Thermo Scientific).

### 4.8. Fluorescence Microscopy

After 6 weeks in the cell culture, the rings were fixed in 4% formaldehyde at RT for 30 min, permeabilized with 0,1% Triton X-100 (G-Biosciences, St. Louis, MI, USA) in DPBS for 5 min at RT, and then rinsed three times with DPBS. The cytoskeleton was stained with Phalloidin-iFluor 488 (abcam, Cambridge, UK) and diluted 1:1000 in DPBS + 1% bovine serum albumin (BSA) for 90 min at RT in the dark. After washing with DPBS, the nuclei were stained using 16 mL of 1 µg/mL Hoechst 34580 (ThermoFischer Scientific, Waltham, MA, USA) in DPBS for 10 min at RT in the dark. After washing with DPBS three times, the rings were microscoped using a Zeiss Axio Observer.Z1 (Zeiss, Jena, Germany).

### 4.9. Scanning Electron Microscopy (SEM)

For scanning electron microscopy, the rings stained positive in the MTT assay (Section 4.6) were cut into 5–7 mm pieces and fixed in 2.5% glutaraldehyde and 4% formaldehyde in a PHEM buffer at pH 6.9 (60 mM PIPES, 25 mM HEPES, 10 mM EGTA, and 4 mM MgSO_4_) for 3 h at RT. Afterwards, the sample was washed (2 × PHEM, 3 × *dest.* H_2_O) and dehydrated in a graded ethanol series (30%, 50%, 70%, 80%, 97%, and 3 × 100%) using a BioWave Pro+ (Ted Pella, Redding, CA, USA) microwave (40 s, 20 °C, and 250 W). The dehydrated samples were dried on a Range CPD7501 critical point dryer (Polaron, London, UK) mounted on a carbon-coated stub and sputter-coated with a 15 nm thick platinum/palladium layer (EM ACE 600, Leica, Wetzlar, Germany). SEM images were acquired using a FIB-SEM (Crossbeam 540, Carl Zeiss, Jena, Germany) with an accelerating voltage of 1.5 kV and a beam current of 2 nA in analytical column mode.

### 4.10. Flow Cytometry

Flow cytometry was performed as previously described [53]. Briefly, after the third passage (cultivation period of app. 3–4 weeks), 6 × 10^5^ cells were incubated with antibodies against CD73 (FITC, clone AD2; ThermoFischer Scientific), CD31 (PE, clone WM59; BioLegend, Fell, Germany), HLA-DR (ECD, clone Immu-357, Beckmann Coulter, Brea, CA, USA), CD44 (APC, clone G44-26; BD Bioscience, Franklin Lakes, NJ, USA), CD34 (APC/Fire 750, clone 581; BioLegend), CD105 (BV421, clone 43A3, BioLegend), CD14 (Pacific Orange, clone MEM-15, Exbio, London, UK), CD90 (BV650, clone 5E10, BioLegend), and CD45 (Alexa Flour 700, clone HI30, BioLegend) for 20 min at 4 °C. The unstained cells were used as a negative control and were mixed with the stained cells in a second step for comparative analyses. Flow cytometry was performed using a CytoFlex and the CytExpert software (both by Beckmann Coulter, Version 2.4.0.28).

### 4.11. Differentiation of Mesenchymal Stromal Cells

Trilineage differentiation of mesenchymal stromal cells with subsequent staining was performed in duplicate as previously described, with the unstimulated cells serving as the negative controls [53]. Briefly, for osteogenic differentiation, 2 × 10^4^ cells were seeded into a 12-well plate and stimulated with DMEM high glucose (PAN Biotech, Aidenbach, Germany) supplemented with 10% FCS, 1% Pen/Strep, 0.1 µM Dexamethasone, 50 µM L-ascorbic acid-2 phosphate, 10 µM β-glycerophosphate, and 5 µg/mL gentamycin (all from Sigma Aldrich). After three weeks, the cells were stained with alizarin red. For chondrogenic differentiation, a high-density culture (0.8 × 10^6^ cells/well) in a 96-well round bottom plate was treated with the chondrogenic medium (StemPro. Chondrogenesis Differentiation Kit, Thermo Fischer) for three weeks and then stained with alcian blue. For adipogenic differentiation, 4 × 10^4^ cells seeded to a 12-well plate were treated with DMEM high glucose supplemented with 10% FCS, 1% Pen/Strep, 1 µM Dexamethasone, 50 µM Indomethacin, 0.5 mM 3-Isobutyl-1-methylxanthine (IBMX), and 10 µg/mL Insulin and stained with oil-red-O (all from Sigma Aldrich).

### 4.12. Biochemical Analysis of the Surgical Site Tissue

Sample preparation: The liquid content of the aspirated surgical site tissue was removed via centrifugation at 2000× *g* for 10 min and collected for further analysis. Samples of approximately 1 g of the solid portion were fresh frozen in liquid nitrogen and stored at −80 °C until use. The cells were isolated from the remaining solid sample as described in Section 4.3 and treated with a red blood cell (RBC) lysis buffer (RBC cell lysis solution 10× Miltenyi Biotec, Gaithersburg, MD, USA) prior to FACS analysis.

Osmolarity, protein, water- and fat content: The water content of the tissue was determined gravimetrically before and after lyophilization. Liquid tissue samples and the patient’s blood was analyzed for electrolytes, total protein, albumin, hemoglobin, and triglyceride content at the central laboratory of the University Hospital. Unsaturated fatty acids were determined after lipid extraction according to Folch [61] using a commercial lipid assay kit (abcam, ab242305). Briefly, 1 g tissue was treated with 20 mL chloroform/methanol (2:1) in an ultrasonic bath for 4 min. After filtering off the solid tissue, the phases were separated and the chloroform phase was evaporated to dryness. Samples and standards were dissolved in dimethyl sulfoxide (DMSO) and hydrolyzed with 18 M sulfuric acid for 10 min at 90 °C. After cooling to 4 °C for 5 min, the vanillin reagent was added and incubated at 37 °C for 15 min. The absorbance was then measured at 540 nm using a Tecan infinite M200pro Reader.

Growth Factor Analysis: To obtain an overview of growth factor distribution in tissue and blood samples, a human growth factor microarray (abcam, ab134002) was used according to the manufacturer’s instructions. Briefly, the membrane was incubated overnight at 4 °C with the sample (250 µg total protein). This was followed by incubations with biotin-conjugated anti-cytokines for 2.5 h at RT and HRP-conjugated streptavidin for 2 h at RT. Then, chemiluminescence was induced with the kit’s detection buffer, measured with an ECL chemocam imager (Intas), and analyzed with Image J (version 1.53c).

ELISAs: The concentrations of VEGF and PDGF growth factors were determined using ELISAs (human duoset ELISA kit, R&D Systems, bio techne) according to the manufacturer’s protocol and measured at 450 nm using a Multiscan Ascent Reader (Thermo Scientific). The liquid component of the sample and the patient´s serum was diluted 1:10. [62]

Statistics: Statistical analysis was performed using R-Studio (version 2022.07.0, Robert Gentleman, Ross Ihaka, University of Auckland, 1992). Continuous variables (patient age, sample weight, MNC number, and generation time) are presented as mean ± standard deviation and categorical variables (gender and MTT-stained biomaterial rings after cultivation) are presented as frequency and percentage. Ordinal parameters (CFU count) are expressed as mean ± standard deviation. For statistical analysis of ATP-assays and ELISAs, the normal distribution of data was validated using Shapiro–Wilk normality test. The results were obtained using a Student’s t-test and linear models (linear mixed model fit via REML). Differences were considered significant at *p* < 0.05.

## 5. Conclusions

Surgical site-released tissue in total arthroplasty harvested under vacuum represent an excellent source for bone regeneration. Both materials, TCP and PCL-TCP, allow for osteogenic differentiation of the cells with advantages for PCL-TCP in vitro. SSRT is a promising source for bone regeneration since it is easy to access and contains multiple osteopromotive agents mimicking the initial tissue and precondition in fracture healing. Clinical data will show if the stimuli of SSRT-augmented bone substitute materials are strong enough to heal delayed or non-unions as well as critical-size bone defects in patients. If so, this strategy will reduce the indication for autologous bone grafting and reduce also the associated co-morbidity to the benefit of the patient. Even if the complex in vivo interactions between cells, cytokines, growth factors, mechanical forces, and scaffolds are poorly understood in detail, the view of Paracelsus, “He who heals is right”, is more relevant than ever.

## Figures and Tables

**Figure 2 ijms-24-15877-f002:**
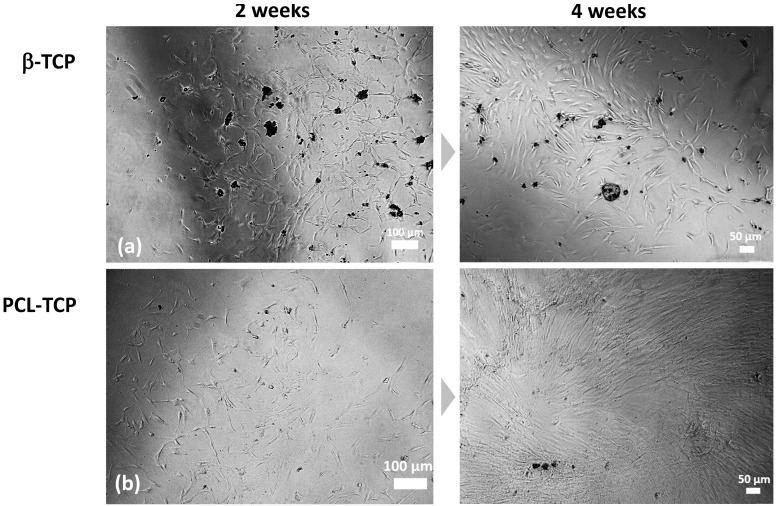
(**a**,**b**) Phase contrast micrographs of adherent MSCs cultured with β-TCP (**a**) and PCL-TCP (**b**) after 2 weeks and 4 weeks in culture.

**Figure 3 ijms-24-15877-f003:**
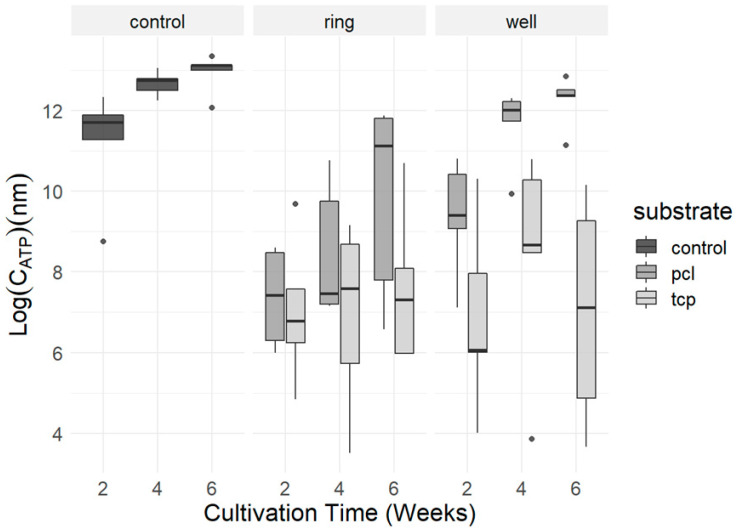
Visualization of ATP concentrations of control, β-TCP, and PCL-TCP conditions compared over the time course of 6 weeks. Ring and well conditions were measured separately. Boxplots indicate the median within the 25–75% percentile. The dots represent single outliners during measurements.

**Figure 4 ijms-24-15877-f004:**
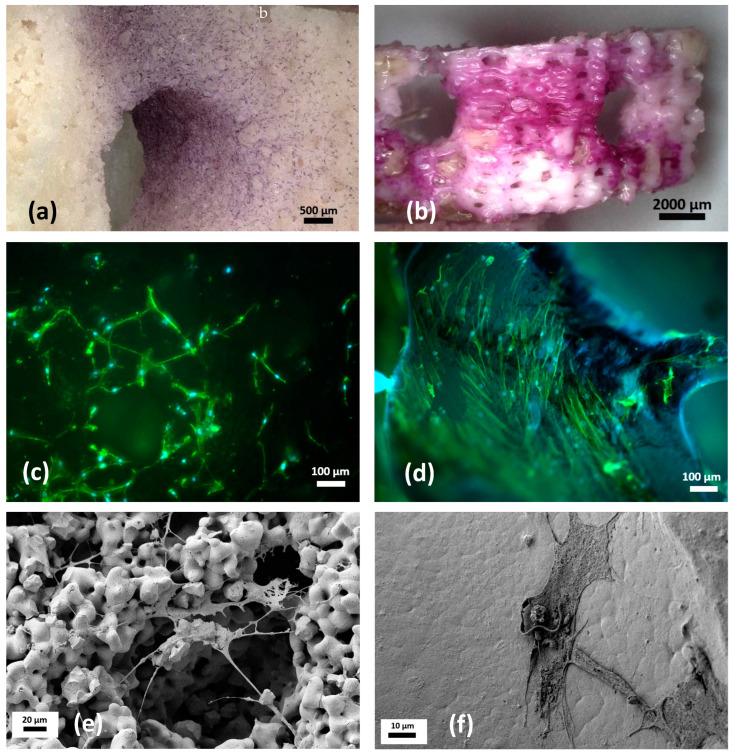
(**a**–**f**) Photomicrographs of viable, purple-stained cells on the β-TCP scaffold (**a**) and the PCL-TCP scaffold (**b**) after MTT staining. The cylindric rings demonstrating a growth of cells appearing from the centre to peripheries. This corresponds also to fluorescence microscopy after nuclei staining. The non-quantitative analysis confirmed the lower cell proliferation rate in TCP (**c**) compared to PCL-TCP (**d**). Cell morphology differed significantly in SEM: in porous TCP specimen, cell processes must overbridge the irregularities in the surfaces leading to thinner morphology and longer stolons. In contrast, PCL-TCP showed flat cuboid cells with some morphological characteristics of an osteoblast and MSCs in cell culture flasks (**f**).

**Figure 5 ijms-24-15877-f005:**
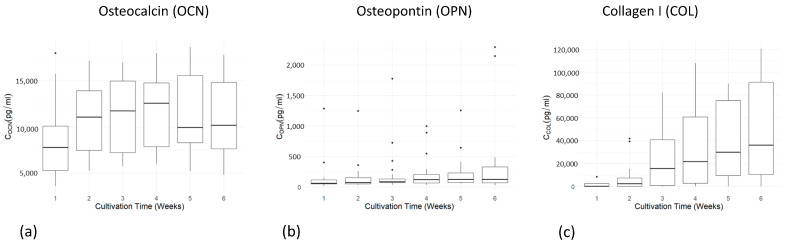
(**a**–**c**) Diagrams of osteocalcin (**a**), osteopontin (**b**), and collagen I (**c**) concentration in cell culture medium over time for 6 weeks. Representation as boxplots indicating the median within the 25–75% percentile. The dots represent single outliners during measurements.

**Figure 6 ijms-24-15877-f006:**
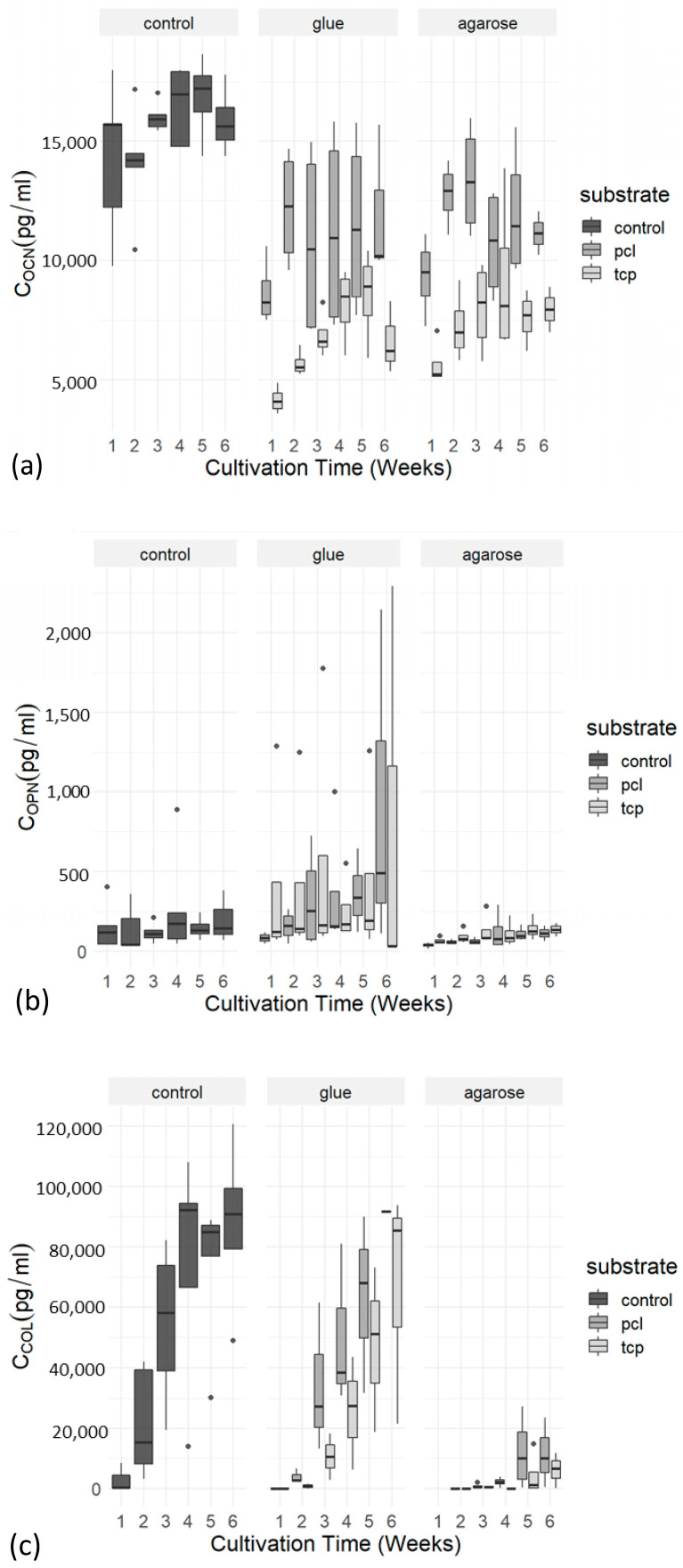
(**a**–**c**) ELISA boxplots of the osteogenic markers osteocalcin (**a**), osteopontin (**b**), and collagen I (**c**). Boxplots compare the concentration of ocn, opn, and col I in the control group with the PCL-TCP and β-TCP group over the cultivation time of 6 weeks. Groups are further subdivided into the condition’s agarose and glue, to examine if the chosen adhesive, used for the attachment of the scaffolds onto the plastic wells, makes a difference in protein expression. The boxblots indicating the median within the 25–75% percentile. The dots represent single outliners during measurements.

**Figure 7 ijms-24-15877-f007:**
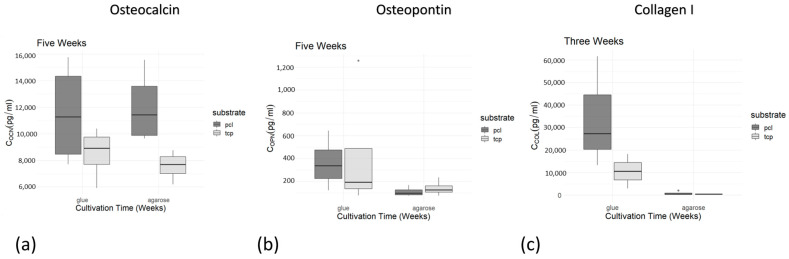
(**a**–**c**) Boxplots of osteocalcin (**a**), osteopontin (**b**), and collagen I (**c**) concentration exemplarily after 5 weeks (for collagen I after 3 weeks) comparing the PCL-TCP and β-TCP group regarding ocn, opn, and col I concentrations. PCL-TCP and β-TCP group are subdivided into subgroups depending on fixing the biomaterial specimen with agarose or glue. Boxplots indicate the median within the 25–75% percentile. The single dots represent single outliners during measurements.

**Figure 8 ijms-24-15877-f008:**
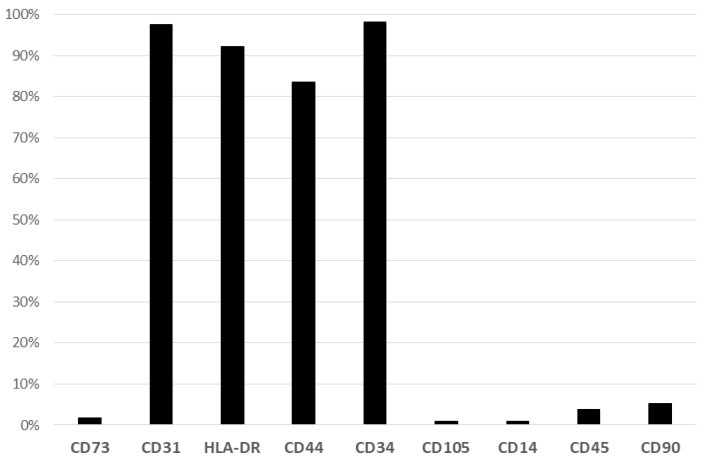
Percentage of surface marker expression of cells isolated from the surgical site tissue (*n* = six patients). Bar chart showing the expression of the antibodies against CD14, CD31, CD34, CD44, CD45, CD90, CD105, and HLA-DR in the collected tissue samples for the biochemical analysis of the aspirated SSRT. Expression of CD31, HLA-DR, CD34, and CD44 are positive.

**Figure 9 ijms-24-15877-f009:**
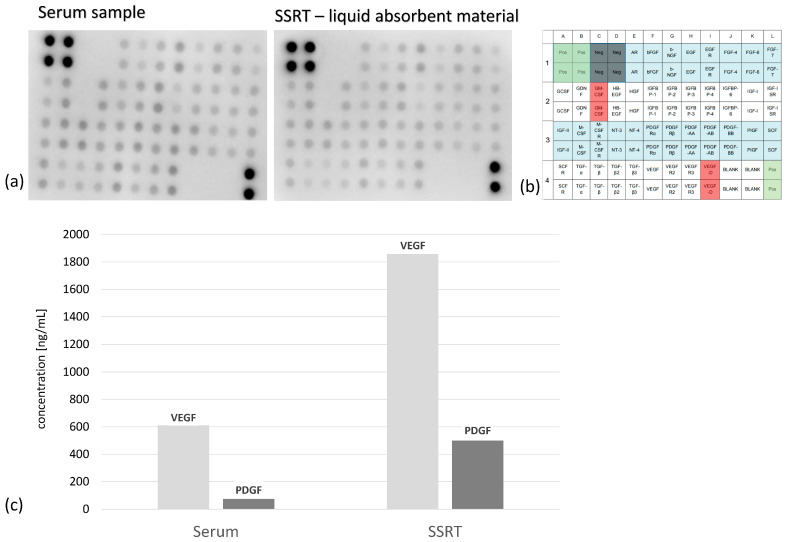
(**a**–**c**) Representative X-ray of the growth factor array from the plasma samples (**a**) and the liquid absorbent material (**b**) of one patient. The right scheme displays the respective coordinate reference number # for analyte identification. Unframed spots are assay-specific reference spots. For VEGF and PDGF, we found significant differences in the concentration in SSRT vs. serum (SD VEGF_serum_: 312.8, SD VEGF_SSRT_: 1018.0, SD PDGF_serum_: 64.7, and SD PDGF_SSRT_: 226.5), *n* = 3.

**Figure 10 ijms-24-15877-f010:**
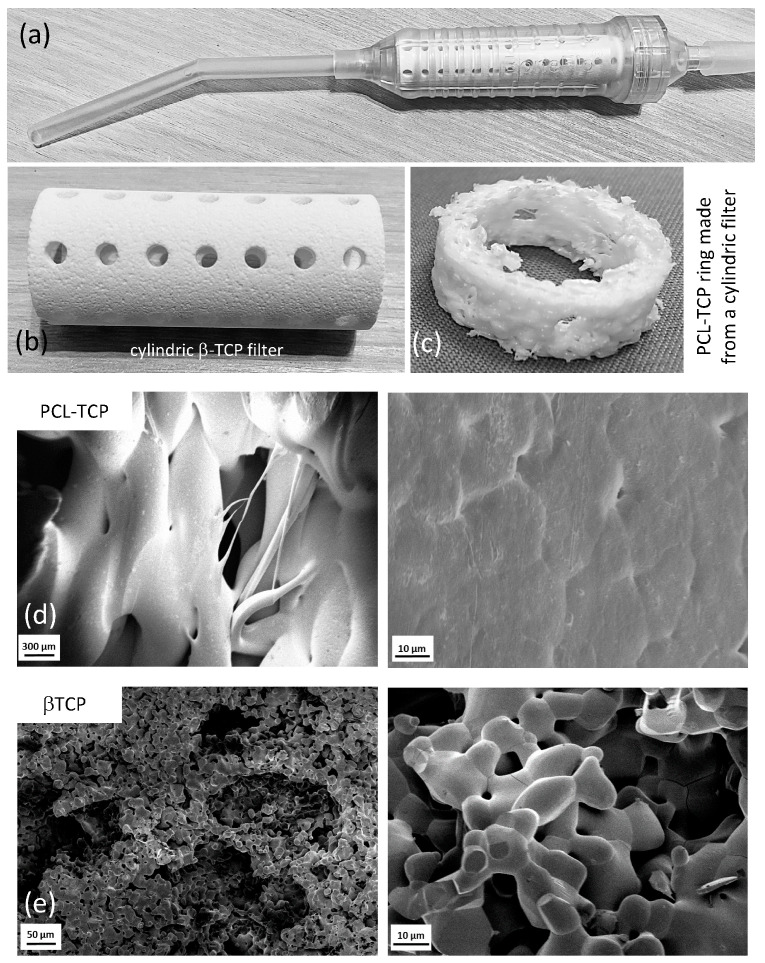
(**a**–**c**) Biomaterial specimen with a cylindric shape designed as an internal tissue filter for a commercial surgical suction handle (**a**). The pores allow the fluid components to pass through, whereas solid pieces of tissue (e.g., bone fragments and small pieces of fat) were retained by the small holes (**b**). For experimental purposes, the cylinders were cut into rings (**c**). Cells isolated from SSRT were cultivated in the middle of the ring. SEM in different magnification showed a typical flat cobblestone surface pattern of PCL-TCP (**d**), whereas the porous β-TCP specimen appeared amorphous and showed multiple interconnecting pores (**e**).

**Table 1 ijms-24-15877-t001:** Selection of typical cytokines and growth factors and their role in bone healing (no claim for completeness).

PDGF	provides fibroblast chemotaxis, improves osteoblast proliferation, modifies extracellular matrix, increases TGF-β release from macrophages
TGF-β	promotes extracellular matrix formation, stimulates collagen and fibronectin synthesis
IGF-I	stimulation of skeletal growth, modification of cells which are involved in tissue regeneration
VEGF	promotion of proliferation and migration of endothelial cells, influence of bone formation and bone healing
IL-1β	protects against infection, activates monocytes
IL-4	regulates immune response, modulates osteoblasts and macrophages, stimulates ECM
IL-6	stimulation of hematopoiesis, signal enhancement in endothelial and immune cells
TNF-α	proliferation and differentiation of different cell types, determinates’ synthesis of ECM, key role in healing and inflammation.

**Table 2 ijms-24-15877-t002:** Mean concentrations of the osteogenic marker proteins osteocalcin, osteopontin, and collagen I in the glue and agarose conditions as determined via ELISA. Student’s *t*-test revealed no statistical significance (*p*-value).

	Mean Group Glue [ng/mL]	Mean Group Agarose [ng/mL]	*p*-Value
Osteocalcin	10.027	9.809	0.8925
Osteopontin	0.395	0.124	0.09402
Collagen I	24.693	0.705	0.07512

**Table 3 ijms-24-15877-t003:** Analysis of electrolytes, proteins, and lipids in the aspirated surgical site tissue. *n* = six patients. Values are listed as mean ± SD.

Electrolytes	Tissue	Blood Samples	Reference Values (Blood)
Sodium	128.8 mmol/L ± 17.2	137.9 mmol/L ± 9.3	135–150 mmol/L
Potassium	>10 mmol/L	4.7 mmol/L ± 0.6	3.5–5 mmol/L
Chloride	111.7 mmol/L ± 14.6	105.7 mmol/L ± 4.1	98–112 mmol/L
Calcium	1.1 mmol/L ± 0.6	2.0 mmol/L ± 0.5	2.3–2.6 mmol/L
Magnesium	0.8 mmol/L ± 0.2	0.8 mmol/L ± 0.1	1.75–4 mg/dL
Phosphate	7.6 mg/dL ± 2.9	4.5 mg/dL ± 1.4	2.6–4.5 mg/dL
Total protein	6.9 g/dL ± 1.7	6.6 g/dL ± 0.9	6–8.4 g/dL
Albumin	4.75 g/dL ± 1.1	4.4 g/dL ± 0.8	3.5–5.5 g/dL
Hemoglobin	10.3 g/dL ± 2.12	11.17 g/dL ± 1.8	12–18 g/dL
Hematocrit	26.6% ± 9%	32% ± 5%	37–50%
Erythrocytes	3.1/pL ± 1.1	3.61/pL ± 0.6	4–5.9/pL
Leukocytes	3.6/nL ± 1.5	6.09/nL ± 1.3	4–10/nL
Platelets	180.3/nL ± 90.9	283.1/nL ± 60.6	150–400/nL
Thrombocrit	0.25% ± 0.0%	0.29% ± 0.07%	0.23–0.24%
Tricglycerides	204.2 mg/dL ± 69.4	162.0 mg/dL ± 62.4	≤150 mg/dL
Unsaturated fatty acids	9.19% ± 2.1%		
Water content	49.3% ± 7.2%		

## Data Availability

The data presented in this study are available on request from the corresponding author. The data are not publicly available due to ethical restrictions.

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
