# Peer review of "Surgical Site-Released Tissue Is Potent to Generate Bone onto TCP and PCL-TCP Scaffolds In Vitro"

_ijms, 2023, doi:10.3390/ijms242115877_

Round 1
Reviewer 1 Report
Comments and Suggestions for Authors
This paper demonstrated that the SSRT in total arthroplasty harvested under vacuum could be an excellent source for bone regeneration onto both TCP and PCL-TCP scaffolds in vitro, where PCL-TCP outperformed TCP. This manuscript can be accepted after minor revision. Please address the comments below.
1. Have the authors considered including pure PCL to compare with TCP and PCL-TCP composite in this study? Will the high hydrophobicity of pure PCL affect the cell adhesion and growth?
2. In the caption of Figure 5. the (e) was missing.
3. Can the authors also include the SEM images of the surface morphology of pristine TCP and PCL-TCP?
Author Response
Reviewer 1
Comments and Suggestions for Authors
This paper demonstrated that the SSRT in total arthroplasty harvested under vacuum could be an excellent source for bone regeneration onto both TCP and PCL-TCP scaffolds in vitro, where PCL-TCP outperformed TCP. This manuscript can be accepted after minor revision. Please address the comments below.
Comment of the reviewer
- Have the authors considered including pure PCL to compare with TCP and PCL-TCP composite in this study? Will the high hydrophobicity of pure PCL affect the cell adhesion and growth?
Answer of the authors
We thank the reviewer for this constructive comment! Of course, the comparison with pure PCL would allow to draw more conclusions of the effect of PCL within the PCL-TCP bone substitute material. However, this would require an extra study group and complete repetition of the investigation since this extra group was not considered in this study. We add this important aspect in the discussion part at the end of the paper.
“In the future, it might be useful to compare also pure PCL as a control, in order to document the PCL-related in vitro effects under the scenario described in this study.”
- In the caption of Figure 5. the (e) was missing.
Answer of the authors
The authors thank for pointing out this formal error and checked the manuscript again. Here, we found that the “e” appeared in the figure and assume that the mistake happened during pdf. file -building. However, we informed the publisher about this problem.
Can the authors also include the SEM images of the surface morphology of pristine TCP and PCL-TCP?
Answer of the authors
We thank the reviewer for this hint and added the required SEM micrographs in the manuscript. These are presented in the revised figure 2d and c. In addition, the figure´s legend was completed: “SEM in different magnification showed a typical flat cobblestone surface pattern of PCL-TCP (d) whereas the porous b-TCP specimen appeared amorphous and showed multiple interconnecting pores (e).”)

Reviewer 2 Report
Comments and Suggestions for Authors
1. The authors in the section "Introduction" consider in detail the participation of various growth factors in the processes of bone tissue regeneration. In the experimental work, quantitative data on growth factors in the samples were obtained, but very little attention is paid to this issue in the "Discussion" section. How can growth factors and implantable scaffolds be interconnected?
2. The authors write that "In this study we investigated if surgical site released tissue components are competent to initiate bone regeneration in vitro when incubated only two different scaffolds qualified as bone void fillers." The authors describe in detail how cells isolated from tissue interact with synthetic scaffolds. How are "surgical site released tissue components competent to initiate bone regeneration"?
3. Line 160-165 The authors write about a way to evaluate the interaction of cells with scaffolds. In the "Results" section there is only the data "Light microscopic evaluation during the culture period" - Figure
3.4. Line 147 – the authors should give a more accurate description of the characteristics of the resulting composite scaffold, as well as specify the characteristics for PCL or provide a link to the relevant publications.
5. Line 305-312 in order to talk about a higher proliferative potential of cells in the presence of scaffolds, it is recommended to present light microscopy data in the initial period of time and after prolonged cultivation. In the process of proliferation, cells can migrate both into the pores of the scaffold and onto the surface of the culture plastic. Since the porosity data is not presented in the article, it is difficult to draw conclusions.
6. The cells were cultured for 6 weeks. Such long-term cultivation is desirable to be carried out in a bioreactor. If there were no such conditions, then it is necessary to take into account that nutrients cannot pass into the small pores of the scaffold. I agree with the data presented in Figure 5, a and b, scaffolds have different pore sizes.
7. In the "Discussion" section, the authors are recommended to justify the conduct of biochemical studies to assess growth factors.
8. The section "Conclusion" is recommended to be expanded and supplemented.
Author Response
Reviewer 2
Comments and Suggestions for Authors
- The authors in the section "Introduction" consider in detail the participation of various growth factors in the processes of bone tissue regeneration. In the experimental work, quantitative data on growth factors in the samples were obtained, but very little attention is paid to this issue in the "Discussion" section. How can growth factors and implantable scaffolds be interconnected?
Answer of the author
We thank the reviewer for this helpful comment and expanded the discussion section in the revised version of the manuscript. The interaction between growth factors and implantable scaffolds is very relevant and a sophisticated topic. Regarding the length of the manuscript and to prevent excessive reports and explanations, we focus on relevant aspects in this field of research and hope that these will meet the requirements of the reviewer expectation.
“The interconnections between scaffolds and growth factors in bone regeneration are sophisticated and subject of current research and scientific discussion. Especially the polymer polycaprolactone (PCL) is a promising candidate to bind growth factors as soluble proteins. This includes not only the ability to use growth factor-loaded PCL scaffolds for a controlled release of proteins [23], e. g. by electrospinning [24] but also the potency of unloaded PCL to bind local growth factors from the adjacent microenvironment such as in cell cultures or body fluids. One example for this phenomenon is VEGF, a factor which is crucial for bone regeneration and neovascularization and showed high levels in our study. Suárez-González et al. incubated PCL scaffolds in modified simulated body fluids and found out that peptide versions of vascular endothelial growth factor (VEGF) and bone morphogenetic protein 2 (BMP2) were bound with efficiencies up to 90% to mineral-coated PCL scaffolds [25]. Moreover, some data suggest that the release of growth factors such as VEGF or PDGF adhered onto PCL is strongly dependent from the solubility and type of the mineral coating [26-30]. In addition, there are several techniques to functionalize PCL-scaffolds for protein binding such as plasma hydrolysis, alkali hydrolysis or aminolysis. [31,32]. We did not further investigate these effects in our study but are aware that further investigations are required to elucidate the interconnections between TCP or PCL-TCP and defined proteins (e. g. VEGF, BMPs or PDGF).”
Comment of the reviewer
- The authors write that "In this study we investigated if surgical site released tissue components are competent to initiate bone regeneration in vitro when incubated only two different scaffolds qualified as bone void fillers." The authors describe in detail how cells isolated from tissue interact with synthetic scaffolds. How are "surgical site released tissue components competent to initiate bone regeneration"?
Answer of the author
The authors thank the reviewer for this critical hint. Since we believe, that not only a single protein but many soluble and eventually also some solid components of the local tissue are required to initiate the well-orchestrated and coordinated pathways of bone formation, the surgical site released tissue during orthopaedic surgery was identified as a potential source for tissue regeneration as described in figure 1. In our study we found some indication which support this hypothesis (components from surgical site released tissue may promote bone formation). However, the answer to the question how „surgical site released tissue components” can initiate bone regeneration is hypothetical and should be addressed stronger in the manuscript as required by the reviewer. Following this helpful comment of the reviewer, we added some possible explanations in the discussion part of the revised manuscript. Moreover, since we did not investigate bone formation in vivo, we used a more careful wording regarding the transferability of in vitro results to the in vivo situation:
“But how can components of the surgical site released tissue initiate bone formation? The answer to this question remains hypothetical. SSRT concentrates the soluble and solid components of mechanical traumatized tissue. It consists of a highly activated biological composite, requesting signal-mediated assistance from different systems of the body (immune system, regeneration system, senso-motoric system) to prevent further damage and loss of tissue, and thus loss of function [53]. These signals derived from a mixture of burst platelets, which release its cytokines and growth factors (a), from factors of the activated coagulation system (b) as well as the activated complement (c) and immune system (d) [54]. Some signals may also be released by ripped tissue fragments (e) such as soft tissue (fat, muscle, fascia, periosteum), as well as bone and its marrow [55-61]. These effects are accompanied by local electrolyte displacement such as increased extracellular K+ due to cellular death and result at least in an aseptic inflammation as shown in figure 1. Since the ability to adhere onto plastic surfaces is a criterion for mesenchymal stroma cells - the progenitors of osteoblasts - the PMMA-filter of the harvesting device may have also a potential to increase the number of MSCs and thus promote osteoblast differentiation. As published previously, cells isolated from SSRT have shown great osteopromotion in vitro [5,13]. This hypothesis is also supported by Groven et al. who used the same tissue collector device (BoneFlo®) and showed exclusively expressed osteogenic miRNAs, as well as overall enhanced osteogenic marker gene expression in SSRT [62].
Therefore, we believe that the application of SSRT into a local bone defect with all its inflammatory and regenerative stimuli is a promising candidate to initiate bone formation, even in non-unions or critical size bone defects. However, we are aware that the transformation of in vitro data to the in vivo situation is limited. Especially, further studies are required to get a more comprehensive quantitative biochemical profile and kinetic of those factors and cytokines which are predominantly involved in bone formation.“
- Line 160-165 The authors write about a way to evaluate the interaction of cells with scaffolds. In the "Results" section there is only the data "Light microscopic evaluation during the culture period"
We thank the reviewer for his valuable comment. The relationship between the microscopic evaluation and the self-selected scoring system for the evaluation was not described precisely enough. We have revised the corresponding section:
“The optical scoring system showed that the β-TCP group exhibited cell growth in 52% of all samples, meaning that in 14 out of 27 samples one of the categories +/++/+++/++++ was fulfilled, and that cell growth was generally occurring in the well. Particularly strong cell growth was observed in the β-TCP wells in 26% of cases, meaning that in 7 out of 27 cases the cell growth could be classified as +++ or ++++. In comparison, the PCL-TCP group showed cell growth in 70% of all samples, so that it could be evaluated that general cell growth, according to category +/ ++/ +++ or ++++ was present in 19 out of 27 samples. Strong cell growth, corresponding to the categories +++ or ++++, was seen in 59% of cases (in 16 out of 27 samples) for the PCL-TCP group.”
Comment of the reviewer
- Line 147 – the authors should give a more accurate description of the characteristics of the resulting composite scaffold, as well as specify the characteristics for PCL or provide a link to the relevant publications.
Answer of the authors
We thank the reviewer for this comment and have complemented the figure 2 showing surface morphology of pristine TCP and PCL-TCP (d, e). Moreover, we contacted the manufacturer of PCL-TCP scaffold and asked for more details regarding the characteristics of the composite scaffold. Unfortunately, we did not receive more information. However, since the material was used also in other in vitro and in vivo studies, we have added the most relevant publications allowing the interested reader to get this information: “Further characteristics of the composite material can be found in the literature [14-17].”
Comment of the reviewer
- Line 305-312 in order to talk about a higher proliferative potential of cells in the presence of scaffolds, it is recommended to present light microscopy data in the initial period of time and after prolonged cultivation. In the process of proliferation, cells can migrate both into the pores of the scaffold and onto the surface of the culture plastic. Since the porosity data is not presented in the article, it is difficult to draw conclusions.
Answer of the authors
Thank you for this critical hint. As suggested, we added also a light microscopy photo showing an earlier (initial) phase of cultivation (please, see figure 3). It is correct that the cells have at least two opportunities – to migrate into the pores or onto the surface of the scaffolds or to stay on the plastic surface of the petri dish or both to some extent. You are right, we are not able to give a detailed information regarding this interaction, although we showed in SEM as well as in fluorescence microscopy that cells were found into the deeper layers of the material. For this reason, we did not overstress the interpretation of this aspect. Moreover, we clarified this aspect in the discussion part of the paper:
“In our study we showed that cultivated cells from SSRT migrated into the 3D micropores of both scaffolds and found qualitative differences in SEM Although, we did not quantify the number of cells adhered to the scaffold directly, we measured the released ATP concentration as indirect evidence for cell migration.”
- The cells were cultured for 6 weeks. Such long-term cultivation is desirable to be carried out in a bioreactor. If there were no such conditions, then it is necessary to take into account that nutrients cannot pass into the small pores of the scaffold. I agree with the data presented in Figure 5, a and b, scaffolds have different pore sizes.
Answer of the authors.
We thank the reviewer for this comment. In our study culture medium was exchanged every 3rd day guaranteeing a sufficient supply by nutrients. However, we are aware that not only insufficient diffusion of nutrients through the 3D network might be a problem, but also the progressive adhesion of protein layers onto the surface may clog and block small pores and thus inhibiting the diffusion of nutrition. We add this aspect into the discussion section of the revised manuscript:
“Moreover, long-term cultures may be problematic for nutrients to supply the cells through the 3D network of the scaffolds by diffusion. Here, also the progressive adhesion of protein layers onto the surface may clog and block small pores and thus inhibiting the diffusion of nutrition. For further studies, these disadvantages may be solved using bioreactors.”
Comment of the reviewer
- In the "Discussion" section, the authors are recommended to justify the conduct of biochemical studies to assess growth factors.
Answer of the author
We refer to this aspect and added a sentence within the discussion part of the revised manuscript:
“Especially, further studies are required to get a more comprehensive quantitative biochemical profile and kinetic of those factors and cytokines which are predominantly involved in bone formation. “
Comment of the reviewer:
- The section "Conclusion" is recommended to be expanded and supplemented.
Answer of the authors:
The authors are following the recommendation of the reviewer and added the following sentences:
“SSRT is a promising source for bone regeneration since it is easy to access and contains multiple osteopromotive agents mimicking the initial tissue and precondition in fracture healing. Clinical data will show if the stimuli of SSRT augmented bone substitute materials are strong enough to heal delayed or non-unions as well as critical size bone defects in patients. If so, this strategy will reduce the indication for autologous bone grafting and reduce also the associated co-morbidity to the benefit of the patient. Even, if the complex in vivo interactions between cells, cytokines, growth factors, mechanical forces and scaffolds are poorly understood in detail, the view of Paracelsus, "He who heals is right", is more relevant than ever.”
PLEASE SEE ALSO ATTACHMENT

Round 2
Reviewer 2 Report
Comments and Suggestions for Authors
All the comments of the reviewer have been taken into account and the manuscript can be accepted for publication.